# Expression of Retroelements in Cervical Cancer and Their Interplay with HPV Infection and Host Gene Expression

**DOI:** 10.3390/cancers13143513

**Published:** 2021-07-14

**Authors:** Gislaine Curty, Albert N. Menezes, Ayslan C. Brant, Miguel de Mulder Rougvie, Miguel Ângelo M. Moreira, Marcelo A. Soares

**Affiliations:** 1Oncovirology Program, Instituto Nacional de Câncer (INCA), Rio de Janeiro 20231-050, Brazil; gcf.science@gmail.com; 2Institute of Cancer and Genomic Sciences, College of Medical and Dental Sciences, University of Birmingham, Birmingham B15 2TT, UK; albertmenezes@gmail.com; 3Genetics Program, Instituto Nacional de Câncer (INCA), Rio de Janeiro 20231-050, Brazil; ayslanbrant@gmail.com (A.C.B.); miguelm@inca.gov.br (M.Â.M.M.); 4Division of Infectious Diseases, Department of Medicine, Weill Cornell Medicine, New York, NY 10065, USA; mid2034@med.cornell.edu

**Keywords:** retroelements, HERV, L1, LINE-1, cervical cancer, HPV

## Abstract

**Simple Summary:**

Retroelements are endogenous DNA elements present in the human genome. They have a key role in tumorigenesis and cancer progression. Several studies have reported their expression as biomarkers and immunotherapeutic targets to cancer. Their expression can be induced by epigenetic alteration and virus infections. Retroelement overexpression has been shown in distinct types of cancer, such as breast cancer, melanoma, renal cell carcinoma and prostate cancer. However, studies on other specific types of cancer, such as in cervical cancer, are scarce. Here we explored the expression of retroelements in cervical cancer as well as their interplay with HPV infection and their association with expression of neighboring genes. We find expression of retroelements specific to cervical cancer, associated with tumor histological type and HPV infection, as a new target as biomarkers and for immune therapy approaches.

**Abstract:**

Retroelements are expressed in diverse types of cancer and are related to tumorigenesis and to cancer progression. We characterized the expression of retroelements in cervical cancer and explored their interplay with HPV infection and their association with expression of neighboring genes. Forty biopsies of invasive cervical carcinoma (squamous cell carcinomas and adenocarcinomas) with genotyped HPV were selected and analyzed for human endogenous retrovirus (HERV) and long interspersed nuclear element 1 (L1) expression through RNA-seq data. We found 8060 retroelements expressed in the samples and a negative correlation of DNA methyltransferase 1 expression with the two most expressed L1 elements. A total of 103 retroelements were found differentially expressed between tumor histological types and between HPV types, including several HERV families (HERV-K, HERV-H, HERV-E, HERV-I and HERV-L). The comparison between HPV mono- and co-infections showed the highest proportion of differentially expressed L1 elements. The location of retroelements affected neighboring gene expression, such as shown for the interleukin-20 gene family. Three HERVs and seven L1 were located close to this gene family and two L1 showed a positive association with *IL20RB* expression. This study describes the expression of retroelements in cervical cancer and shows their association with HPV status and host gene expression.

## 1. Introduction

Cervical cancer is the fourth most common cancer as well as the fourth leading cause of cancer death among women worldwide [1]. Human papillomavirus (HPV) infection is a hallmark of the vast majority of cervical cancer cases. HPV belongs to the Papillomaviridae family and is composed of double-stranded DNA genome of approximately 8 kb. Over 200 HPV genotypes have already been identified, yet only approximately 40 of them are able to infect the genital tract [2,3]. These are classified into high-risk, potentially high-risk or low-risk genotypes for cervical cancer, based on their carcinogenic properties and epidemiologic studies. High-risk HPV are 16, 18, 31, 33, 35, 39, 45, 51, 52, 56, 58, 59 and 68, while the potentially high-risk types are 53, 66, 70, 73, and 82. HPV16 and 18 are the most frequent high-risk types, accounting for approximately 70% of all invasive cervical cancer cases worldwide [4,5,6].

HPV infects the cervical epithelium through microabrasions of the single layered squamous cellular junction between the endo- and ectocervix [7]. Upon infection, HPV persists as an episomal DNA genome in the nucleus of the infected cell, which encodes all their viral proteins (E1, E2, E4, E5, E6, E7, E8, L1 and L2). During tumoral progression, the HPV episomal genome can suffer disruption and can be randomly integrated into the cellular genome. This event leads to dysregulation of viral and host gene expression, stimulation of cellular growth, inhibition of cellular differentiation and induction of chromosomal instability [8,9,10,11,12,13].

Epigenetic dysregulation related to cancer has been associated with expression of genes involved in tumorigenesis and cancer progression, such as retrotransposons expression [14,15,16,17,18]. Retrotransposons are a class of endogenous elements found in all eukaryotic organisms, but not in prokaryotes [19]. They compose about 42% of the human genome and are divided in two groups, long terminal repeats (LTR) and non-LTR elements [20,21]. Human endogenous retroviruses (HERV) are LTR elements and make up about 8% of the human genome; while the long interspersed nuclear elements-1 (L1) are non-LTR elements and compose about 17% of the human genome [20]. The expression of these retroelements has been related to different types of cancer, such as melanoma, breast cancer and prostate cancer [14,22,23]. However, while many studies have reported retrotransposon expression for some types of cancer, such assessment lacks for other types, such as cervical cancer. Thus, in the current study we characterized HERV and L1 expression and analyzed their expression in relation to HPV type infection and tumor histological types of cervical cancer.

## 2. Materials and Methods

### 2.1. Samples

Forty biopsies of invasive cervical carcinoma (26 squamous cell carcinomas, 11 adenocarcinoma and three other histological types) were collected from patients at Instituto Nacional de Câncer (Rio de Janeiro, Brazil) before cancer treatment and stored in RNAlater (QIAGEN, Chatsworth, CA, USA) at −80 °C. The median age of patients was 44 years (21–74 years). All samples were HPV genotyped and multiple and single infections, which were described previously [24,25]. Twenty samples were positive for HPV16 infection, twelve for HPV18 infection and eight samples showed other HPV types, such as HPV39, HPV31, HPV 33 and HPV45. Twenty-two samples were infected by a single HPV genotype and 18 samples showed multiple HPV genotypes as previously described [24,25].

### 2.2. RNA-Seq

All samples were processed for RNA extraction with the QIAGEN Allprep DNA/RNA mini kit (QIAGEN) according to the manufacturer’s instructions. Total RNA (0.1 to 4 μg) was used for library preparation with TruSeq RNA Sample Prep Kit (Illumina, San Diego, CA, USA) according to the manufacturer’s recommendations. Libraries were sequenced using a 2× 101-bp paired-end sequencing strategy in a HiSeq 2500 platform (Illumina), as previously reported [24,25].

### 2.3. Host Gene and Retrotransposon Expression Analyses

The demultiplexed reads in fastq format were obtained with Casava 1.8. Reads were filtered by quality mean phred score (>20) and length (>30 nucleotides) using PRINSEQ [26]. Filtered reads were aligned to the reference human genome (hg38) using Bowtie2 and its output was used in HTSeq-Count to calculate the host gene expression, and in Telescope software to define and quantify retroelements (HERV and L1) expression [27,28,29]. Although we are aware that assignment of the correct genomic source of “younger” TE transcripts, such as some non-LTR, TE is a challenging task, Telescope is a generally accepted algorithm used for unambiguous locus assignment [27]. Retroelements that were present in less than two samples were removed from the analysis. The model based on the negative binominal distribution, DESeq2, was used to calculate differentially expressed retroelements comparing HPV16 vs. HPV18, HPV-coinfected vs. HPV-monoinfected and adenocarcinoma vs. squamous cell carcinoma using Wald-test [30]. HERV and L1 with adjusted *p*-value < 0.01 and log2FoldChange > 1.0 were considered differentially expressed and results were shown using pheatmap and ggplot R packages. Gene expression was normalized by internal Deseq2 size factors and plotted using ggplot R package. In addition, differentially expressed HERV and L1 genes according to HPV type and tumor histological type data were visualized in Venn plots using the Venn Diagram R package.

We localized retroelements to nearby HPV integration sites through the integrative genomics viewer (IGV) software (Broad Institute, Cambridge, MA) using HPV integration site data, human genome (*GRCh38*) annotation and retroelement annotation. Five upstream and five downstream retroelements expressed nearby HPV integration sites were initially considered, but only retroelements that expressed at least one sample were evaluated with respect to their expression in relation to the nearby HPV integration site. Retroelement expression was plotted using ggplot2 R package. We also obtained data of the closest host genes to each DE retrotransposon (ranging from 0―when the element is within the gene―to 21,752,076 pb) through their human genome localization using Gencode v. 31 and retrotransposon annotations. Gene set enrichment analysis (GSEA) was performed with WebGestalt (available at http://webgestalt.org/, accessed on 20 November 2020) using Reactome as the functional pathway database to analyze the biological function of host genes neighboring retroelements [31]. Correlation analysis between retroelements and host gene expression using linear regression and scatter plots were performed using ggplot2 R packages.

## 3. Results

### 3.1. Retrotransposon Expression in Cervical Cancer

We explored retroelements (HERV and L1) gene expression in cervical cancer. A total of 8060 HERV and L1 were found expressed in the samples. The most expressed HERV and L1 among the samples were *L1FLnI_11p15.4m*, *L1FLnI_20q13.12e*, *HARLEQUIN_Yq11.221c*, *L1FLnI_16p11.2c*, *L1FLnI_8q21.11w*, *L1FLnI_5q35.1c*, *HERVH_13q33.3*, *HERVH_20p11.23b*, *HERVH_1p31.3d* and *ERV316A3_6p21.33c* (Appendix A). These elements represented 53% of all HERV and L1 expressed in the cervical cancer samples analyzed.

We also analyzed the association between expression of genes involved in the DNA methylation mechanism, such as those of the DNA methyltransferase family (DNMT1, DNMT3A, DNMT3B), and the most expressed retroelement loci, since DNA methylation is a cellular mechanism accounting for the control of retrotransposon expression (Figure 1, Appendix A). We found a weak but significant negative correlation of *DNMT1* expression with *L1FLnI_5q35c* (R = −0.38, *p* = 0.016) and *L1FLnI_11p15.4m* (R = −0.34, *p* = 0.031) expression (Figure 1F,H), which is congruent to the hypothesis of a relationship between the low level of *DNMT1* expression and LINE expression.

### 3.2. Retrotransposons Are Differentially Expressed between Cervical Cancer Histological Types and HPV Genotype Infections

We performed retroelement differential gene expression analysis for squamous cell carcinoma vs. adenocarcinoma, HPV18 vs. HPV16 and HPV coinfections (HPVco) vs. HPV monoinfections (HPVmono). We found a total of 103 retroelements differentially expressed; 68 (32 L1 (47%) and 36 HERV (53%)) between squamous cell carcinoma and adenocarcinoma, 17 (seven L1 (41%) and 10 HERV (59%)) between HPV18 and HPV16; and 18 (14 L1 (78%) and 4 HERV (22%)) between (HPVco) and HPVmono (Figure 2A,B; Appendix A). The comparison between HPV coinfections and HPV monoinfections showed a higher proportion of L1 differentially expressed than the remaining comparisons (Figure 2B). Different HERV families were also identified in the analysis (Figure 2B), such as HERV-K and HERV-H (Figure 2 and Appendix A). The comparison of retroelement loci differential expressions to HPV infections, histological types and HPV types showed 10, 58 and 15 exclusive retroelement loci, respectively (Figure 2C). Moreover, differentially expressed retroelement loci were clustered into three distinct groups with respect to those comparisons (Appendix A).

### 3.3. Expression of Retroelements and Nearby Host Genes

HPV is able to affect gene expression by random integration in the human genome in tumor cells. Therefore, we analyzed HERV and L1 expression close to HPV integration sites (Appendix A). We found variable results regarding the expression levels of the nearby retroelements when compared to cases where no HPV integration occurred at the respective genome location, and expression levels also did not appear to be influenced by the orientation of the HPV relative to those of the retroelement loci (Figure 3). Our data indicate that HPV integration at the 11p15.1, 13q22.1, 20q13.13 and 22p13.2 genome locations did not affect nearby retroelements expression.

The differential expression of retroelements can influence the expression of neighboring host genes. Thus, we also analyzed their expression in cervical cancer. Two hundred forty-nine retroelement loci were found closest to 164 host genes (from 0 to 21,752,076 pb). We also analyzed host gene biological functions by GSEA (pathway, reactome) and gene pathways were enriched for several gene sets (Table 1). However, only the interleukin-20 family signaling showed a significant adjusted *p*-value (*q*-value = 0.0012) for that enrichment (Table 1).

Three HERVs and seven L1 were localized close to the interleukin-20 family signaling genes (*IL19*, *IL20*, *IL20RA*, *IL20RB*, *IL22RA2*) (Appendix A). We analyzed the association between retroelements expression and interleukin-20 gene family expression (Figure 4). Linear regression analysis showed a positive relation between *L1FLnI_3q22.3h* (R = 0.982, *p* < 0.001) and *L1FLnI_3q22.3k* (R = 0.5767, *p* < 0.001) with *IL20RB* expression (Figure 4F,H).

## 4. Discussion

Expression of retroelements is related to cancer, and their contribution to tumorigenesis and cancer progression has been reported [14,15,16,32]. In the current study we characterized retroelement expression profiles in cervical cancer as well as their complex interaction with HPV infection and expression of neighboring host genes. We found five L1 and HARLEQUIN, HERV-H and ERV316A3 families commonly expressed in the samples analyzed. HERV expression of HERV-H and HARLEQUIN families has been reported in head and neck cancer [33]. It is well known that retroelement expression is active in cancer cells, germ line cells and during embryogenesis, but in non-malignant cells their expression is epigenetically suppressed [14,16,34]. We found expression of one specific HARLEQUIN locus, HARLEQUIN_Yq11.221c, which is located in the Y chromosome, an unexpected finding in cervical cancer cells. Telescope, used for retroelement identification, shows lower false positive rates (below 0.1%) when compared to other algorithms designed for the same purpose [27]. We think this is an assignment error specific to cervical cancer, since we did not detect expression of this locus in tumor cells of other cancers, such as breast, head and neck cancer [33,35]. Also, we found a high identity of HARLEQUIN_Yq11.221c sequence to another element in chromosome 3 (data not shown). These data highlight the importance of studying HERV expression in cervical cancer to discover the expression of novel retroelements and putative biomarkers in cervical cancer.

DNA methylation plays an important role in the control of retroelement expression. The latter are reported to be hypomethylated and overexpressed in cancer, which has been associated with poor cancer prognosis [14,16,34,36]. L1 hypomethylation level increases from normal cervical tissue to cervical cancer tissue and is associated with cervical cancer progression [37]. Herein, we found a weak but significant negative correlation of DNA methyltransferase 1 (*DNMT1*) expression to *L1FLnI_5q35* (R = −0.38, *p* = 0.016) and *L1FLnI_11p15.4m* (R = −0.34, *p* = 0.031) expression. The DNA methyltransferase family is composed of enzymes responsible for catalyzing the transfer of methyl groups from S-adenosyl methionine to DNA. Their overexpression is reported to lead the transcriptional repression of several genes through hypermethylation in diverse kinds of cancer [38]. Moreover, disruption in the *DNMT1* gene by CRISPR-Cas9 editing in human neural progenitor cells upregulates L1 expression [39]. Together, these data suggest that the level of *DNMT1* expression also can affect LINE expression in cervical cancer. The weak, yet significant correlations found, are compatible with the notion that multiple epigenetic mechanisms for silencing retroelements exist in mammalian cells, including the action of enzymes of the apolipoprotein B editing catalytic (APOBEC) family and of other innate sensors that target nucleic acids of viral origin, in addition to methylation [40].

A total of 103 differentially expressed retroelement loci were identified in this study, including HERV-K. HERV-K is the most studied HERV family and its expression has been reported in different kinds of cancer, including cervical cancer [14,41]. HPV infection is the main factor related to cervical cancer development. However, the interplay of this virus with retroelement expression remains to be explored in cervical cancer. HPV is able to integrate randomly in the human genome during cancer progression, which may affect host gene expression [13,42,43]. Our data show that HPV integration at 11p15.1, 13q22.1, 20q13.13 and 22p13.2 loci does not appear to affect retroelement expression. Several factors may explain such lack of association, including the genetic distance of the HPV integration site to the retroelements. The distance observed from the integrated HPV to the retroelements in the cases analyzed herein was from 20 kb to 2.118 kb. Even the shortest distance observed among our cases (20 kb) may be distant enough to preclude observation of any interference in expression. Moreover, the limited number of samples studied with respect to similar HPV integration sites makes the study of influence in expression anecdotal and precludes any statistical analysis. Thus, additional studies are necessary to evaluate the influence of HPV integration on retroelement expression.

On their turn, the localization of upregulated retroelements in the human genome could also affect neighbor host genes [14,44,45,46,47]. We found three HERVs and seven L1 localized close to the interleukin-20 family signaling genes (*IL19*, *IL20*, *IL20RA*, *IL20RB*, *IL22RA2*). Members of the IL20 family mediate the communication between epithelial cells and leukocytes, induce proinflammatory cytokine and chemokine production, and stimulate the proliferation of epithelial cells [48]. The linear regression analysis showed a positive correlation between *L1FLnI_3q22.3h* (R = 0.982, *p* < 0.001) and *L1FLnI_3q22.3k* (R = 0.5767, *p* < 0.001) with the *IL20RB* expression. IL20RB composes IL19, IL20 and IL24 receptors. Overexpression of IL20RB has been correlated with cell invasion and migration enhancement, cell proliferation, and poor prognosis in renal cell carcinoma [49]. These data could suggest that combined L1 and IL20 family gene expression may induce cellular signaling pathway alterations, leading to abnormal cellular proliferation and increased cellular invasion capacity of the tumor.

In summary, our findings describe retroelement expression profiles in cervical cancer and their association with endogenous factors, such as *IL20* family genes and *DNMT1*, which showed a significant correlation with up- and downregulation of L1 expression, respectively. In addition, HPV infection, as an exogenous factor, can also modulate retroelement expression, suggesting a complex interaction and a multifactorial relationship between cervical cancer and exogenous and endogenous factors.

## Figures and Tables

**Figure 1 cancers-13-03513-f001:**
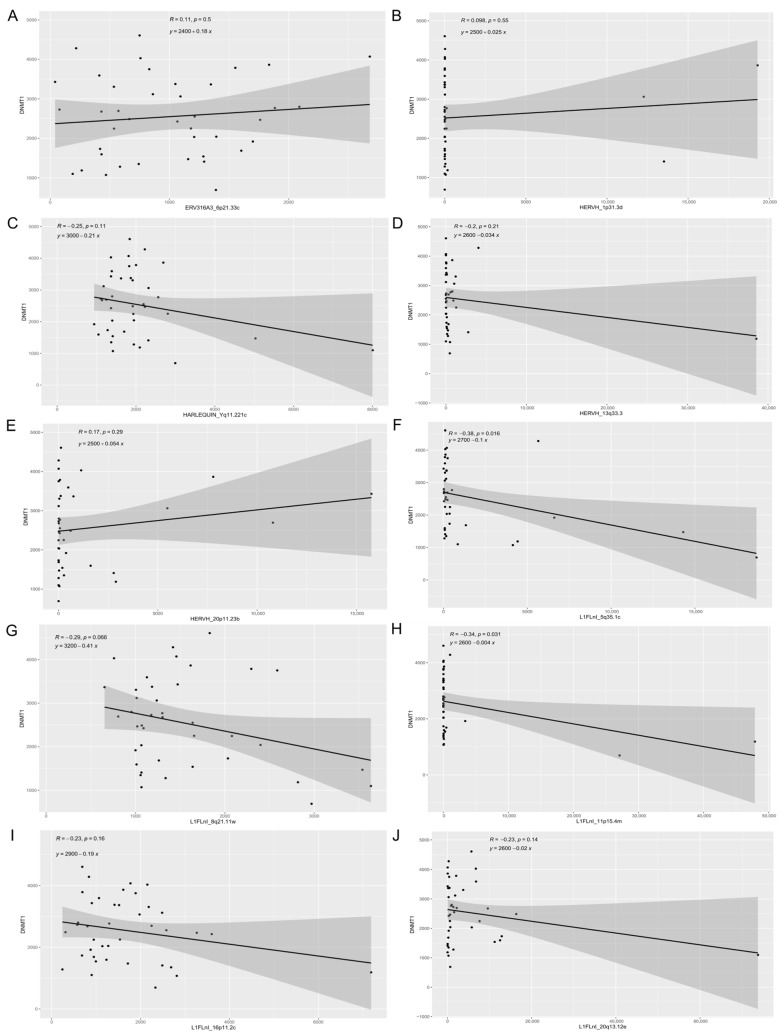
Correlation between DNA methyltransferase 1 (*DNMT1*) and retroelement expression. Scatter plots show correlation of DNMT1 gene expression with (**A**) *ERV316A3_6p21.33c*, (**B**) *HERVH_1p31.3d*, (**C**) *HARLEQUIN_Yq11.221c*, (**D**) *HERVH_13q33.3*, (**E**) *HERVH_20p11.23b*, (**F**) *L1FLnI_5q35.1c*, (**G**) *L1FLnI_8q21.11w*, (**H**) *L1FLnI_11p15.4m*, (**I**) *L1FLnI_16p11.2c* and (**J**) *L1FLnI_20q13.12e*. Pearson correlation coefficient (R), *p*-value (p) and linear equation are shown.

**Figure 2 cancers-13-03513-f002:**
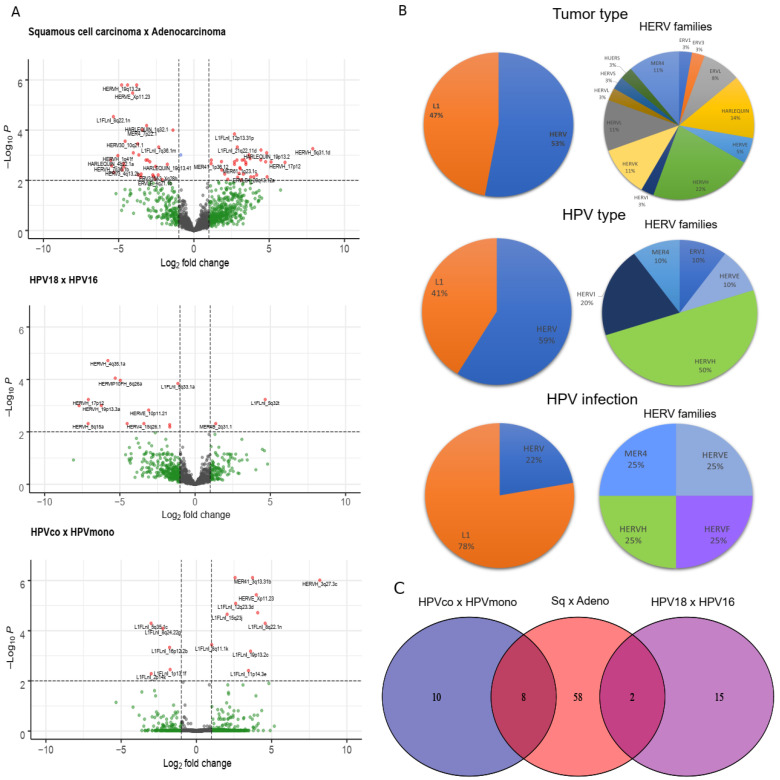
Retrotransposon loci differentially expressed in cervical cancer. (**A**) Volcano plot shows differentially L1 and HERV loci expressed with respect to cervical tumor histological type (squamous cell carcinoma x adenocarcinoma), infecting HPV type (HPV18 vs. HPV16) and HPV infection (HPV coinfection (HPVco) vs. HPV monoinfection (HPVmono)). (**B**) Proportion of differentially expressed L1 and HERV found in samples (left panels) and of HERV families found (right panels) with respect to each comparison depicted in A. (**C**) Venn plot shows exclusive and common HERV and L1 loci found in each analysis (squamous cell carcinoma (Sq) vs. adenocarcinoma (Adeno), HPV18 vs. HPV16 and HPVco vs. HPVmono).

**Figure 3 cancers-13-03513-f003:**
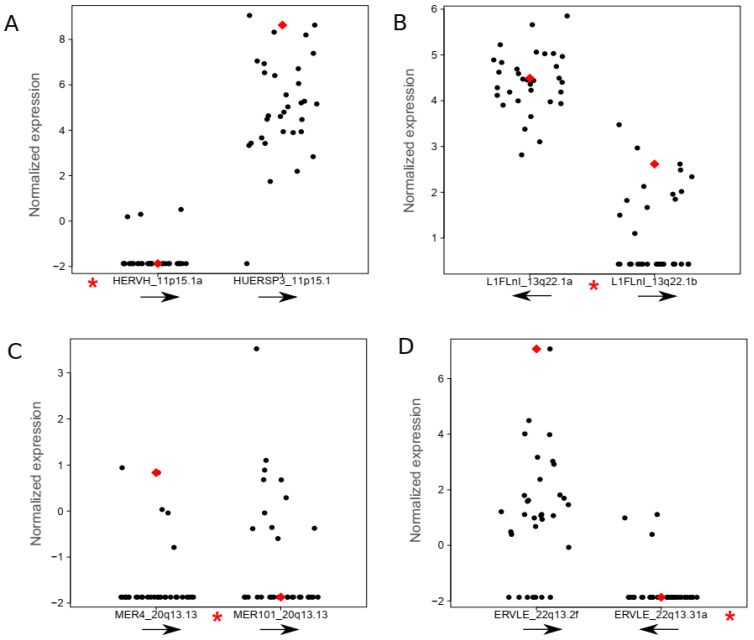
Retroelement expression near HPV integration sites. The normalized expression for the two nearest retroelements to the HPV integration site is shown in plots. (**A**) *HERV-H* and *HUERSP3*, (**B**) *L1Fnl_13q22.1a* and *L1Fnl_13q22.1b*, (**C**) *MER4* and *MER101*, (**D**) *ERVLE_22q13.31f* and *ERVLE_22q13.31a*. Red dots represent the sample in which HPV is integrated in each case, while black dots depict the remaining samples without HPV integration. Red asterisks in the *x*-axis of each panel show the location of HPV integration relative to the two closest retroelements. All HPV are integrated in an antisense DNA orientation. Black arrows below each retroelement show their relative DNA orientation.

**Figure 4 cancers-13-03513-f004:**
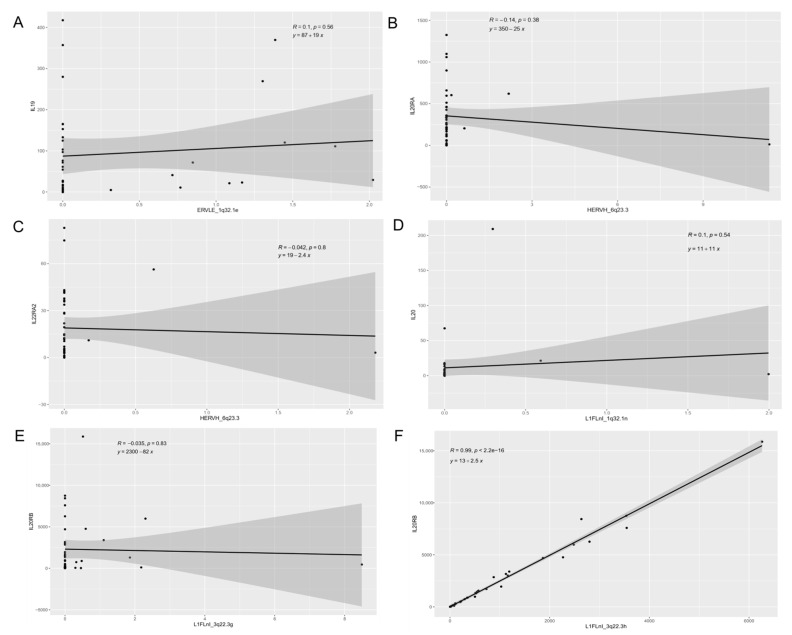
Interleukin-20 gene family and retroelement expression. Scatter plots show (**A**) *IL-19* x *ERVLE_1q32.1e*, (**B**) *IL20RA* vs. *HERVH_6q23.3*, (**C**) *IL20RA2* vs. *HERVH_6q23.3*, (**D**) *IL20* vs. *L1FLnI_1q32.1n*, (**E**) *IL20RB* vs. *L1FLnI_3q22.3g*, (**F**) *IL20RB* vs. *L1FLnI_3q22.3h*, (**G**) *IL20RB* vs. *L1FLnI_3q22.3i*, (**H**) *IL20RB* vs. *L1FLnI_3q22.3k*, (**I**) *IL20RB* vs. *L1FLnI_3q22.3l* and (**J**) *IL19* vs. *L1FLnI_1q32.1n*. Pearson correlation coefficient (R), *p*-value (p) and linear equation are shown.

**Table 1 cancers-13-03513-t001:** Gene Set Enrichment Analysis (GSEA) from nearby host genes of retrotransposons.

Gene Set	Description	*p*-Value	*q*-Value
R-HSA-854691	Interleukin-20 family signaling	7.0383 × 10^−7^	0.0012162
R-HSA-419408	Lysosphingolipid and LPA receptors	0.00011395	0.098454
R-HSA-799990	Metal sequestration by antimicrobial proteins	0.0007144	0.35011
R-HSA-162588	Budding and maturation of HIV virion	0.0010607	0.35011
R-HSA-420029	Tight junction interactions	0.0010607	0.35011
R-HSA-941332	RUNX2 regulates genes involved in cell migration	0.0013215	0.35011
R-HSA-917729	Endosomal Sorting Complex Required For Transport (ESCRT)	0.0014183	0.35011
R-HSA-803205	TP53 signaling	0.0041793	0.75655
R-HSA-803157	Antimicrobial peptides	0.0047203	0.75655
R-HSA-162123	Synthesis of Prostaglandins (PG) and Thromboxanes (TX)	0.0048005	0.75655

## Data Availability

Dataset used in this study is available in GEO (http://www.ncbi.nlm.nih.gov/geo, accessed on 1 December 2020) with access number GSE91065 and GSE144293.

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
