# Peer review of "Expression of Retroelements in Cervical Cancer and Their Interplay with HPV Infection and Host Gene Expression"

_cancers, 2021, doi:10.3390/cancers13143513_

Round 1
Reviewer 1 Report
This paper investigates potential relationships between retroelement expression and HPV infection in cervical cancer. Although the topic is interesting and there are other studies suggesting interplay between viral infection and retroelement activity, this study has major flaws. The main problem, which pervades all of the experiments, is that RNA-seq reads (read length is not stated but typically is 50 bp paired-end) are claimed to be mapped to individual retroelement loci. The software used is Telescope, which has previously been applied to HERVs but not LINE-1 elements. There is no benchmarking or orthagonal validation done to confirm that the detected reads are correctly mapped to a specific source element. Approaching this question by first performing subfamily-wide analysis of retroelement expression would inspire much more confidence--older, more diverged retroelement copies can be mapped to with much greater accuracy than younger copies, but it is impossible to discern from the manuscript the ages and subfamilies of the specific elements being detected. In general, the statistical analyses correlating retroelement expression with DNMTs and adjacent cellular genes are unconvincing and unclear.
Author Response
Transposable elements (TE) are repetitive sequences and present a challenge to RNA-seq dataset analysis because of sequence similarity, leading to uncertainty in its mapping and quantification. Many algorithms have appeared to approach that question, such as Telescope, RepEnrich, TEtranscripts, SalmonTE, among others (https://tehub.org/en/resources/repeat_tools) for RNA-seq and others for Chip-seq. All these algorithms overcome the mappeability issue using different strategies and Telescope has shown a confinable resolution for precisely estimates of TE expression at a loci level. In short, Telescope solves the problem of ambiguously aligned fragments by assigning each sequenced fragment to its most likely transcript of origin. It uses a Bayesian mixture model to represent transcript proportions, unobserved source templates and estimates model parameters using an expectation-maximization algorithm. Through this, Telescope is able to accurately quantify the expression of transposable elements (HERV and LINE-1) in RNA-seq datasets as well as localize TE expression in the exact chromosomal location. A higher-level analysis of TEexpression by TE family suggested by the Reviewer had already been conducted in the original manuscript, and the results presented in Figure 2 and Suppl. Table 1.
We should mention that the HiSeq 2500 apparatus and the Illumina kits used in our experiments generates 101-bp single reads, which after the paired-end runs conducted generates ˜200-bp fragments, not 50-bp as stated by the Reviewer. This sequence length allows more unambiguous assignment of transcripts to the correct locus in the genome. This statement indicating the fragment size of the reads has been highlighted in this revised version of the manuscript.
The validation of Telescope’s algorithm has already been conducted in its original description (https://doi.org/10.1371/journal.pcbi.1006453), and it is not within the scope of this manuscript. However, it should be mentioned that over 20 recent publications, between experimental and review papers, cited or used Telescope. While the reviews usually mention Telescope as an algorithm with improved capability of assigning unambiguous TE transcripts to the right loci, experimental studies have used it with success, even for studying the expression of LINES and SINES, as was the case for one study (https://doi.org/10.1101/2021.04.08.439009).
Despite the improved capacity of Telescope compared to other algorithms, we recognize that assigning the correct source of “younger” TE transcripts is a challenging generalized task in the field, and we added such limitation in the Methods of the revised manuscript.
With respect to the statistical analyses correlating retroelement expression with DNMTs and adjacent cellular genes, we used standard methods for assessing correlations, i.e. linear regression with Pearson’s correlation and associated p-values, considering those < 0.05 as significant.
Reviewer 2 Report
In the manuscript submitted by Curty et al, the authors report the identification of retroelements in cervical cancer by RNAseq analysis. They characterize its association with epigenetic regulators and host gene expression. They detail the differential presentation of retroelements between histological subtypes and frequency of HPV infection. This report is recommended for publication and will garner attention for further in-depth investigation into this topic.
Author Response
We are happy that the Reviewer liked the study and thank him for his contribution.
Reviewer 3 Report
This report is significant because it provides information about associations between retroelements and HPV in cervical cancer that provides new clues about the etiology of this disease. I have the following concerns about the writeup:
The statement on lines 42 and 43 "HPV infection is a necessary, but not a sufficient cause for cervical cancer development" should be modified to incorporate the existence of cervical cancers that have undetected HPV.
The sentence on lines 69 through 71 should have a citation, unless one of the three references in the preceding sentence specifically addresses cervical cancer.
The summary sentence on lines 288-290 is too strongly stated based on the evidence provided in the manuscript. The word "interaction" should be changed to "association" or "correlation" to accurately reflect the level of the data presented.
Author Response
We thank the Reviewer for the contributions made to improve the text of the manuscript, and we have changed all accordingly.
Lines 42-3: we changed the text to “Human papillomavirus (HPV) infection is a hallmark of the vast majority of cervical cancer cases”.
Lines 69-71: Indeed, there are no studies of retroelement expression in cervical cancer, and no reference exists to be cited. We have corrected the text of the sentence to express that.
Lines 288-290: As per the Reviewer’s suggestion we have changed “interaction” to “association”.
Round 2
Reviewer 1 Report
Thank you for addressing original criticisms.
Author Response
Thank you for comments, that were addressed previously.